# Evaluation of the effect of reinforced education on the satisfaction of patients undergoing colonoscopy: A randomized controlled trial

**Negin Farid** [1]*, **Shakila Sharifian**[2], **Raziyeh Ghafouri**[3]

1 Student Research Committee, Department of Medical Surgical Nursing, School of Nursing & Midwifery, Shahid Beheshti University of Medical Sciences, Tehran, Iran, 2 Student Research Committee, School of Nursing & Midwifery, Shahid Beheshti University of Medical Sciences, Tehran, Iran, 3 Department of Medical and Surgical, School of Nursing & Midwifery, Shahid Beheshti University of Medical Sciences, Tehran, Iran

* neginfarid.79@gmail.com, n.farid@sbmu.ac.ir

## Abstract

### Introduction

Fourteen million colonoscopies are performed annually in the United States, with the results dependent on bowel preparation before the procedure. However, suitable preparation is neglected in 20–25% of cases, resulting in increased loss of time, failure to diagnose, procedure repetition, and decreased patient satisfaction. Consequently, the current study was performed to evaluate the effect of reinforced education (RE) on the satisfaction of patients undergoing colonoscopy.

### Methods

This study employed an experimental research method. Participants included 83 patients referred to Ayatollah Taleghani Medical Educational Center for colonoscopy, who were divided into control and intervention groups. The control group received standard education, whereas the intervention group received instruction via media and virtually, and preparation was followed up on the day before the colonoscopy. The research tool was a demographic and satisfaction questionnaire developed by the researcher. Analyses were conducted using IBM SPSS software (v. 20).

### Results

Eighty-three patients, including 47 men and 36 women with an average age of 49.19 years, participated in the study. Mean (standard deviation) patient satisfaction was 11.78 (4.65) in the intervention group and 9.04 (2.95) in the control group; the independent t-test revealed a significant difference between the two groups (P<0.001).

**Data Availability Statement:** All relevant data are within the manuscript and its Supporting information files.

**Funding:** The author(s) received no specific funding for this work.

**Competing interests:** The authors have declared that no competing interests exist.

**Abbreviations:** 3D, Three Dimensions; CVI, Content Validity Index; CVR, Content Validity Ratio; RE, Reinforced Education.

## Conclusions

The study suggests that reinforced education using media and virtual means is effective in the satisfaction of patients undergoing colonoscopy.

## Introduction

Fourteen million colonoscopies are performed annually in the United States [1], with the results dependent on bowel preparation before the procedure [1, 2]. Inadequate preparation fails to diagnose 42% of adenomas and 27% of advanced adenomas. In contrast, suitable colon preparation facilitates a more thorough examination of the mucosa, the presence of polyps and adenomas, and early cancer diagnosis [1]. In 20–25% of cases, however, adequate preparation is neglected [3], resulting in an increased loss of time, failure to diagnose, an increase in cost [3, 4], and a higher risk of procedural adverse events, such as bleeding or perforation [4].

Although proper bowel preparation is the most important factor in colonoscopy [3], patients frequently find them uncomfortable [5]. One in five individuals fails to take the prescribed dose of colonoscopy preparation drugs [1]. In general, gender, constipation, drug use, previous colon surgery, diabetes, obesity, neurological disease, and lack of knowledge influence inadequate preparation [3]. Accordingly, examining the patient's needs and educating them on the significance of colonoscopy preparation is one of the most significant focus areas [3]. Consequently, effective education can increase a patient's willingness to comply with instructions [1].

Education is provided through various methods, such as pamphlets, animations, messages, and videos, which increase patient readiness by 11–36% [2]. Education will be more effective if multiple instructional strategies are utilized [6]. Furthermore, the training will be more effective if it is based on problem-solving techniques, the immediate needs of the patient, and an appreciation of its significance [7].

Due to the significance of patient compliance and bowel preparation in colonoscopy procedures, training can be effective if it is patient-centered [8], interactive [5], individually focused based on the patient's needs [3], and emphasizes the importance of bowel preparation [9]. Furthermore, patient acceptance is improved by all educational methods, including oral and written education [3], face-to-face education, and media-based education [10]. However, new methods such as utilizing technology, media [1], web applications [11], 3D online programs [12], and mobile applications [13, 14] may increase patient acceptance to a greater extent, necessitating additional research on their efficacy [1]. For example, media use can make reinforced education (RE) more effective from admission until the colonoscopy day [15].

The level of patient satisfaction is a crucial criterion for evaluating the quality of service [16]. Patient satisfaction is an indicator of the quality of care, which is a crucial factor in selecting a treatment center; therefore, it is essential to conduct a study to determine how to improve the quality of services and patient satisfaction [17]. To this end, further research must be conducted to boost care recipients' satisfaction [18, 19]. Patient dissatisfaction is caused by insufficient preparation, incorrect diagnosis, and repeated colonoscopies, which can be remedied through proper training and follow-up. Based on the information presented above, it is possible to conclude that RE can improve patient satisfaction and bowel preparation. As a result, the purpose of this study was to assess the impact of RE on the satisfaction of colonoscopy patients.

## Methods

This semi-experimental study evaluated the impact of using media and virtual means to educate Ayatollah Taleghani Medical Educational Center-referred colonoscopy patients. This medical center performs approximately 2,500 colonoscopies per year or close to 200 per month. The study's statistical population consisted of colonoscopy patients referred to the Ayatollah Taleghani Educational Medical Center, where the study sample comprised patients who met the inclusion criteria.

The study was registered with the Iranian Clinical Trial Registry (IRCT20210131050189N5). The authors confirm that all ongoing and related trials for this intervention are registered.

Notably, due to the nature of this study's intervention (education), we were only satisfied with acquiring the code of ethics at the outset. We realized the necessity of registering our protocol with the trials registration center only after submitting the manuscript in PLOS ONE and conducting a more in-depth examination of the World Health Organization (WHO) guidelines. Therefore, retrospectively and by obtaining the necessary permits, the trial was successfully registered.

### Sample size/power

The number of samples in each group was determined by the following equation:

$$n \geq 2 \frac{\left(z_{\alpha/2} + z_\beta\right)^2 \sigma^2}{(\mu_1 - \mu_2)}$$

Probability of type 1 error: $\alpha = 0.05 \rightarrow z_{\alpha/2} = 1.96$
Probability of type 2 error: $\beta = 0.10 \rightarrow z_\beta = 1.28$
Power of observed effect size: $(\mu_1 - \mu_2)/\sigma = 0.75$
Minimum required sample: $n = 2(1.96 + 1.28)^2 \left(\frac{1}{0.75}\right)^2 = 37$

Each group required a minimum of 37 participants. Considering a potential 40% decline in population, the sample size for each group consisted of 50 patients.

### Data collection and sampling

Sampling was conducted between May 22 to June 22, and data collection was completed through August 7. The inclusion criteria were the absence of movement, hearing, and vision disorders, a medical history free of cognitive diseases and depression, the non-use of neuroleptics, the absence of a recent tragedy such as the death of a loved one or hospitalization, and access to online resources and educational videos. Patients who experienced a new tragedy during the research or refused to continue cooperating were excluded.

### Randomization and blinding

Red and blue colored cards were used to divide the participants into the control and intervention groups (Fig 1). To clarify we must explain that our team was stationed in the colonoscopy department of the hospital for 30 consecutive days. Before starting the sampling, the person in charge of reception and head nurse of the department was asked if the patients' visits are based on a certain order? (e.g. age, type of disease, degree of disease). They said that it is only based on the time of each patient's visit to get an appointment and there is no special order. So the team leader prepared an equal number of red and blue cards, shuffled them and put them in sealed envelopes. A number was randomly written on each envelope. These envelopes were

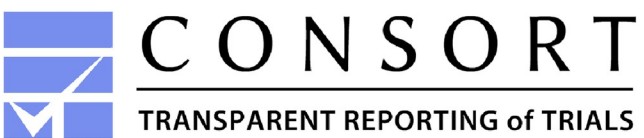

## CONSORT 2010 Flow Diagram

**Enrollment**

Assessed for eligibility (n=120)

Excluded  (n=20)
- Not meeting inclusion criteria (n=8)
- Declined to participate (n=12)
- Other reasons (n=0)

Randomized (n=100)

**Allocation**

Allocated to intervention (n=50)
- Received allocated intervention (n=50)
- Did not receive allocated intervention (n=0)

Allocated to control (n=50)
- Received allocated measures (n=50)
- Did not receive allocated intervention (n=0)

**Follow-Up**

Lost to follow-up (death, out of reach) (n=6)

Discontinued intervention (failure to refer for colonoscopy) (n=1)

Lost to follow-up (death, out of reach) (n=10)

Discontinued intervention (n=0)

**Analysis**

Analysed  (n=43)
- Excluded from analysis (n=0)

Analysed  (n=40)
- Excluded from analysis (n=0)

**Fig 1. CONSORT flowchart for the study.**

provided to the instructor. After fixing the appointment, the patient referred to us as the person in charge of education, According to the order of referral, each patient was given an envelope. If there is a blue card, the patient will enter the control group, and if there is a red card, the patient will enter the intervention group. We did not give the envelope to the patient to open, so that if the patients met each other on the day of the test, they would not be able to share information. Therefore, the opening of the envelope was done by the instructor. In addition, in all stages of sampling and data collection, a person outside the study team who was trained was stationed to monitor the correct process of doing the work.

In the control group, the patient's contact number was obtained and the educational routine of the hospital was implemented. But for the intervention group, we were taking numbers, and 3 days before the test, the training was sent online in the manner mentioned below.

*RG* designed the allocation sequence, *NF* and *SS* enrolled participants, and *NF* and *RG* assigned interventions to each group (*capital letters indicate the authors' initials*). Moreover, patients were blinded to their group classification. In addition, the post-intervention data collector was unaware of participant assignment to control and intervention groups. The two groups were homogeneous, where no statistically significant differences were observed in age, sex, education level, underlying disease, or reason for admission (P<0.05).

## Data collection tool

The data collection tool included a demographic information questionnaire with 20 questions, including the patient's age, sex, level of education, marital status, occupation, type of health insurance, the reason for hospitalization, length of hospitalization, illness history, underlying disease, medical family history, previous colonoscopy experience, place of residence, diet (daily fruit consumption, weekly fast food consumption), ability to perform daily activities, and amount of exercise per day.

The colonoscopy satisfaction instrument was a researcher-made instrument with 12 questions in 3 dimensions (pre-colonoscopy education, colonoscopy experience, and colonoscopy result). A review of the relevant literature was used to develop the tool's components prior to its creation [20–22]. Then, duplicate items were eliminated, categorized, and summarized by the expert panel consisting of ten gastroenterologists and endoscopy nursing experts. Next, face validity (qualitative), content validity (quantitative and qualitative), and construct validity were examined. Ten nurses were requested to provide corrections regarding the use of proper grammar, vocabulary, expression placement, and the inclusion of sufficient evidence to verify the accuracy of the content. The content validity index (CVI) and content validity ratio (CVR) were then determined and validated. The relationship between the items was subsequently determined and summarized using exploratory factor analysis into three factors (pre-colonoscopy education, colonoscopy experience, and colonoscopy result). Cronbach's alpha was used to evaluate instrument reliability; its value was 0.83.

The colonoscopy patients were contacted one week after the procedure, and the aforementioned patient satisfaction questionnaire was administered.

## Intervention

The standard training program was provided to the control group, which received a printed training sheet. The control group received a training sheet on the day of the colonoscopy appointment (the same as the intervention group, but a typed version); however, no follow-up was provided to answer questions or clarify ambiguous situations. In the intervention group, patients were contacted the day before their colonoscopy, and training materials, including

video clips, an application, and an Instagram page, were sent to them via the WhatsApp application, and their questions were answered.

Bisacodyl tablets and Pidrolax powder were utilized for bowel preparation. The morning and afternoon versions of the standard protocol for bowel are detailed below (as recommended by ESGE [24]).

**Colonoscopy appointment between 8:00 to 13:00.** One day before a colonoscopy, only liquids, including water, sugar water, compote water, tea, strained fruit juice (clear and pulp-free), meat broth, and chicken broth, may be consumed beginning at 9 a.m. Avoid soups, dairy products, juices containing fruit particles, compotes, and solid foods such as (bread, rice, potatoes, meat, and ash).

Maintain a fluid-only diet throughout the day. Use one tablet of Bisacodyl at noon, 4 p.m., and 8 p.m.

Use six colonoscopy preparation powders (Pidrolox) between noon and midnight. Each powder is mixed with one liter of water (four large glasses), and one glass is consumed every half-hour. At least three to four hours of walking and physical activity are required daily.

Start fasting in the morning on the day of the test.

**Colonoscopy appointment between 13:00 to 19:00.** One day before the colonoscopy, only liquids, including water, sugar water, compote water, tea, strained fruit juice (clear and pulp-free), meat broth, and chicken broth, may be consumed beginning at 9:00 a.m. Avoid soups, dairy products, fruit juices containing fruit particles, compotes, and solid foods such as (bread, rice, potatoes, meat, and ash).

Maintain a fluid-only diet throughout the day. Use one tablet of Bisacodyl at noon, 4:00 p. m., and 8:00 p.m.

Use five powders for colonoscopy preparation between 4:00 p.m. and 12:00 a.m. Pour each powder into one liter of water (four large glasses) and consume one glass every 20 minutes.

At least three to four hours of walking and physical activity are required daily.

At 8:00 a.m. on the day of the colonoscopy, a preparation powder should be mixed with one liter of water (four large glasses) and consumed. Fasting begins at 10 a.m.

## Data analysis

Analyses were conducted using IBM SPSS software (v. 20, IBM Corp., Armonk, NY). The Kolmogorov-Smirnov tests were used to assess the normality of the distribution of variables. A p-value $< 0.05$ was deemed statistically significant. Either the t-test or the chi-square test was utilized to analyze demographic data. If the data distribution was normal, the paired t-test and student t-test were used to compare variables within and between groups. Notably, we used the 2010 version of the CONSORT reporting guidelines for this manuscript [23].

**Ethical consideration.** The Ethics Committee of the Student Research Committee at Shahid Beheshti University of Medical Sciences approved the study protocol (IR.SBMU.RETECH. REC. 1400.1082) entitled "Evaluation of the effect of reinforced education with media on patient education on colonoscopy satisfaction." To adhere to ethical considerations, the research objectives and methods were explained to the participants, they were assured of the anonymity and confidentiality of their information, and each participant provided written informed consent. They participated voluntarily and had the option to withdraw at any time. This research was registered with IRCT ID IRCT20210131050189N5 in the Iranian Registry of Clinical Trials.

Notably, our intervention involved cleaning the intestine and detecting colorectal pathology early, which was in the patient's best interest. No other interventions were administered.

The liquid diet was safe for the patient due to its nutrients and short duration. Pidrolox powder had no side effects or complications, except for the taste, which was adjusted by adding fragrant citrus juice.

## Results

In this study, 83 patients with an average age of 49.19 years participated, including 47 men and 36 women. Table 1 shows the demographic characteristics of the intervention group (43 individuals) and the control group (40 individuals).

Initially, we examined the homogeneity of the two groups and observed no significant difference (P>0.05). Mean (standard deviation) patient satisfaction was 11.78 (4.65) in the intervention group and 9.04 (2.95) in the control group. The independent t-test revealed a statistically significant difference between the two groups (P<0.001). In the colonoscopy experience factor, the t-test revealed that the difference was not statistically significant (P>0.05). In the colonoscopy results (P = 0.05) and pre-colonoscopy education, the t-test indicated that the difference was statistically significant (P<0.001). The results of patient satisfaction factors between the control and intervention groups are shown in Table 2.

## Discussion

The present study aimed to evaluate the effect of RE on the level of satisfaction of patients undergoing colonoscopy. The results indicated that RE using media and virtual means improved the level of satisfaction of these patients (P<0.01). Guo et al. observed similar effects of RE on bowl preparation [9]. Furthermore, Aggarwal et al. investigated the effect of training on safety, stated that effective training methods should be utilized with regard to patient safety, and emphasized the significance of ongoing training [24]. Bott et al. compared the impact of implementing a specialized training program after treating cardiac arrhythmia to standard

**Table 1. Participants' demographic characteristics.**

| | | Group | | | | Result of Homogeneity |
|---|---|---|---|---|---|---|
| | | Control | | Intervention | | |
| | | Count | N % | Count | N % | |
| Gender | Female | 18 | 21.7% | 18 | 21.7% | $\chi^2$ = 0.083 df = 1 P = 0.773 |
| | Male | 25 | 30.1% | 22 | 26.5% | |
| Marital Status | Single | 7 | 8.6% | 6 | 7.4% | $\chi^2$ = 0.004 df = 1 P = 0.952 |
| | Marriage | 36 | 44.4% | 32 | 39.5% | |
| Education | School | 17 | 21.5% | 15 | 19.0% | $\chi^2$ = 1.703 df = 3 P = 0.636 |
| | High School | 19 | 24.1% | 16 | 20.3% | |
| | Bachelor | 2 | 2.5% | 4 | 5.1% | |
| | Upper | 2 | 2.5% | 4 | 5.1% | |
| Number Colonoscopy | First | 29 | 34.9% | 19 | 22.9% | $\chi^2$ = 5.248 df = 2 P = 0.072 |
| | Second | 10 | 12.0% | 10 | 12.0% | |
| | More | 4 | 4.8% | 11 | 13.3% | |
| Hospitalization | Outpatient | 33 | 39.8% | 35 | 42.2% | $\chi^2$ = 1.619 df = 1 P = 0.203 |
| | Inpatient | 10 | 12.0% | 5 | 6.0% | |
| Family History | No | 33 | 40.2% | 29 | 35.4% | $\chi^2$ = 0.063 df = 1 P = 0.802 |
| | Yes | 10 | 12.2% | 10 | 12.2% | |

**Table 2. Comparative analysis of patient satisfaction between the control and intervention groups.**

|  | Control | | Intervention | | T | P value |
|---|---|---|---|---|---|---|
|  | Mean | Standard Deviation | Mean | Standard Deviation |  |  |
| pre-colonoscopy education | 1.66 | 1.68 | 3.55 | 2.51 | -3.99 | 0.00 |
| colonoscopy experience | 4.57 | 2.17 | 4.95 | 2.13 | -0.68 | 0.49 |
| colonoscopy result | 1.30 | 0.66 | 1.63 | 0.77 | -1.99 | 0.05 |

discharge instructions in their study. According to the study's findings, the training provided by the nurse to the patient 24 to 72 hours after discharge increased patient satisfaction and reduced readmission rates [25]. The findings of Bott et al. and the present research are consistent.

In a study conducted in the radiology department, Lutjeboer et al. demonstrated that promoting self-care can increase patient satisfaction [26]. In the present study, patient-centered training increased patient satisfaction. Similarly, Ezzat et al. found that reducing complications during cardiac catheterization can increase patient satisfaction [17]. Since the reduction of complications is a component of self-care training, their results are also consistent with those of the present study and indicate a rise in satisfaction following the promotion of self-care with training and follow-up of care and preparations.

Patient education is crucial for promoting self-management and decreasing hospitalization and treatment costs [25]. Therefore, costs are reduced by three to four dollars for every dollar spent on patient education. According to presented statistics, the United States spends between 69 and 100 million dollars annually on medical issues caused by a lack of education [26].

The majority of individuals possess cell phones today [27]. Athilingam (2016) reported that there are 7 billion cell phones worldwide and that nearly 95% of the global population uses cell phones [28]. The owner's cell phone is with them throughout the day [28], and its use is beneficial for teaching self-care and disease prevention programs [29]. Utilization and accessibility have increased satisfaction [30]. The current study used mobile and media programs to improve patient education and follow-up. Mobile education in the reinforced educational program was more acceptable and increased patient satisfaction.

Even though our study group was limited to patients undergoing colonoscopy, and this study was conducted with random samples from an Iranian population, and should allude to future studies with much larger sample sizes collected from other geographical areas to confirm the efficacy of the RE procedure, it appears that the result can be generalized to other procedures where the background knowledge of the patient and their family is incomplete and limited, because it reaffirms the importance of obtaining the patient's and family's perspective.

## Research limitations

The outbreak of COVID-19, which hindered data collection, was one of the study's limitations, but it did not affect our findings or generalizability. Neither did the virtual method or telephone follow-up. One of the strengths was the use of several teaching methods.

## Conclusions

Based on the findings of this study, it can be concluded that through media and cyberspace utilization, RE is effective in the satisfaction of patients undergoing colonoscopy. Notably, the current findings are based solely on random samples derived from the Iranian population, and

future studies with much larger sample sizes collected from other geographic regions are required to confirm the efficacy of the RE method.

## Supporting information

**S1 Protocol. Study protocol.**
(DOCX)

**S1 Checklist. Reporting checklist for randomised trial.**
(DOCX)

## Acknowledgments

We would like to thank all participants and those who assisted us in conducting the research, particularly the entire staff of the colonoscopy department at Ayatollah Taleghani Educational Medical Center and the Gastroenterology and Liver Diseases Research Center at the Research Institute for Gastroenterology and Liver Diseases of Shahid Beheshti University of Sciences.

This study is related to the project NO.1400/4584 from Student Research Committee, Shahid Beheshti University of Medical Sciences, Tehran, Iran.

We also appreciate the "Student Research Committee" and "Research & Technology Chancellor" in Shahid Beheshti University of Medical Sciences for their support of this study.

## Author Contributions

**Conceptualization:** Raziyeh Ghafouri.

**Data curation:** Negin Farid, Raziyeh Ghafouri.

**Formal analysis:** Negin Farid, Shakila Sharifian, Raziyeh Ghafouri.

**Investigation:** Negin Farid.

**Methodology:** Negin Farid, Shakila Sharifian.

**Validation:** Raziyeh Ghafouri.

**Writing – original draft:** Negin Farid.

**Writing – review & editing:** Negin Farid, Shakila Sharifian, Raziyeh Ghafouri.

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
