## [Decision Letter · Decision Letter 0]

7 May 2023

PONE-D-23-02398Evaluation of the Effect of Reinforced Education on the Satisfaction of Patients Undergoing Colonoscopy: a Randomized TrialPLOS ONE

Dear Dr. Farid,

Thank you for submitting your manuscript to PLOS ONE. After careful consideration, we feel that it has merit but does not fully meet PLOS ONE’s publication criteria as it currently stands. Therefore, we invite you to submit a revised version of the manuscript that addresses the points raised during the review process.

We look forward to receiving your revised manuscript.

Kind regards,

Antonio Brillantino

Academic Editor

PLOS ONE

Journal Requirements:

2. Thank you for submitting your clinical trial to PLOS ONE and for providing the name of the registry and the registration number. The information in the registry entry suggests that your trial was registered after patient recruitment began. PLOS ONE strongly encourages authors to register all trials before recruiting the first participant in a study.

        1) your reasons for your delay in registering this study (after enrolment of participants started);

        2) confirmation that all related trials are registered by stating: “The authors confirm that all ongoing and related trials for this drug/intervention are registered

3. We note that you have selected “Clinical Trial” as your article type. PLOS ONE requires that all clinical trials are registered in an appropriate registry (the WHO list of approved registries is at "https://www.who.int/clinical-trials-registry-platform/network/primary-registries" https://www.who.int/clinical-trials-registry-platform/network/primary-registries"  https://www.who.int/clinical-trials-registry-platform/network/primary-registries and more information on trial registration is at http://www.icmje.org/about-icmje/faqs/clinical-trials-registration/). Please state the name of the registry and the registration number (e.g. ISRCTN or ClinicalTrials.gov) in the submission data and on the title page of your manuscript. a) Please provide the complete date range for participant recruitment and follow-up in the methods section of your manuscript. b) If you have not yet registered your trial in an appropriate registry, we now require you to do so and will need confirmation of the trial registry number before we can pass your paper to the next stage of review. Please include in the Methods section of your paper your reasons for not registering this study before enrolment of participants started. Please confirm that all related trials are registered by stating: “The authors confirm that all ongoing and related trials for this drug/intervention are registered”. Please see http://journals.plos.org/plosone/s/submission-guidelines#loc-clinical-trials for our policies on clinical trials.

4. We note that the original protocol that you have uploaded as a Supporting Information file contains an institutional logo. As this logo is likely copyrighted, we ask that you please remove it from this file and upload an updated version upon resubmission.

Additional Editor Comments:

According with reviewers comments, the manuscript needs a a careful and thorough review to be considered for acceptance.

Reviewers' comments:

Reviewer's Responses to Questions

**Comments to the Author**

1. Is the manuscript technically sound, and do the data support the conclusions?

Reviewer #1: No

Reviewer #2: Partly

2. Has the statistical analysis been performed appropriately and rigorously? 

Reviewer #1: No

Reviewer #2: No

3. Have the authors made all data underlying the findings in their manuscript fully available?

Reviewer #1: Yes

Reviewer #2: Yes

4. Is the manuscript presented in an intelligible fashion and written in standard English?

Reviewer #1: Yes

Reviewer #2: Yes

5. Review Comments to the Author

Reviewer #1: Dear authors,

good luck with your efforts. However, I think that the scientific writing of the article is not enough. My suggestions are attached on the file. It can be developed in line with the recommendations. I wish you good work.

Reviewer #2: This manuscript presents data analysis from a randomized control trial (RCT) to compare the effectiveness of the reinforced education (RE) procedure (intervention), versus control, on improving satisfaction of colonoscopy patients. The topic is of importance, the study was registered as a RCT (with a valid number), and was approved by the respective IRB/Ethics Committee. While the study objectives sound interesting, is important, and on target, some shortcomings were observed, in regards to abiding by the CONSORT guidelines for conducting and reporting results of high-quality randomized controlled trials (RCTs). Some other (statistical) comments were also provided.

1. Methods:

Methods reporting need some work. An orderly manner is suggested, following CONSORT guidelines, without repeating information, such as Trial Design, Participant Eligibility Criteria and settings, Interventions, Outcomes, sample size/power considerations, Interim analysis and stopping rules, Randomization (details on random number generation, allocation concealment, implementation), Blinding issues, etc, should be mentioned. The authors are advised to create separate subsections for each of the possible topics (whichever necessary), and that way produce a very clear writeup. They are advised to write it carefully, following nice examples in the manuscript below:

https://www.sciencedirect.com/science/article/pii/S0889540619300010

Specific comments:

(a) For instance, the randomization and allocation concealment should be made very clear (they are NOT the same thing); the trial staff recruiting patients should NOT have the randomization list. Randomization should be prepared by the trial statistician, and he/she would not participate in the recruiting.

(b) The manuscript does not provide details on the randomization procedure; any reasoning, why a block randomization was not used, which is often recommended to ensure a balance in sample size across groups?

https://www.ncbi.nlm.nih.gov/pmc/articles/PMC2267325/

(c) The authors write in the Abstract: "In order to prevent the dissemination of information between the two groups, first the data of the control group was collected". This is not clear! You may easily attain information blockade between the 2 groups via blinding, and a proper randomization procedure. More details needed.

(d) Sample size/power: A sample size/power statement is made available, but its recommended to place it as a separate subsection within the Methods section, following CONSORT guidelines.

(e) Statistical Analysis:

(e1) Alternative analysis methods under data non-Gaussianity is not mentioned, such as using well-known nonparametric tests.

(e2) Several additional covariates (via the questions) were also obtained. Why a formal regression analysis was not conducted?

(e3) It's not clear if data was collected longitudinally; please provide details. If that is the case, mixed modeling should be used.

(e4) Overall writeup has various inconsistencies; please fix incomplete sentences. For example, a sentence started like "it should be mentioned......" (Page 7). Sentences start with Capital letters.

2. Results & Conclusions:

(a) The authors should check that any statement of significance should be followed by a p-value in the entire Results section. Otherwise, the Results section look OK.

(b) Conclusions should state that the current findings are ONLY based on the random samples derived from an Iranian population, and should allude to future studies with much larger sample sizes and collected at other geographical areas to confirm the effectiveness of the RE procedure.

6. PLOS authors have the option to publish the peer review history of their article (what does this mean?). If published, this will include your full peer review and any attached files.

Reviewer #1: No

Reviewer #2: No

---

## [Author Response · Author response to Decision Letter 0]

17 Jul 2023

Thank you for giving us the opportunity to submit a revised draft of our manuscript titled [Evaluation of the effect of reinforced education on the satisfaction of the patients undergoing colonoscopy: a randomized controlled trial] to PLOS ONE. We appreciate the time and effort that you and the reviewers have dedicated to providing your valuable feedback in our manuscript. We are grateful to the reviewers for their insightful comments on our paper. We have been able to incorporate changes to reflect most of the suggestions provided by the reviewers.We have highlighted the changes within the manuscript.

I will thank you in advance to see the track change file and also the Response to Reviewer.

---

## [Decision Letter · Decision Letter 1]

25 Sep 2023

PONE-D-23-02398R1Evaluation of the effect of reinforced education on the satisfaction of patients undergoing colonoscopy: a randomized controlled trialPLOS ONE

Dear Dr. Farid,

Thank you for submitting your manuscript to PLOS ONE. After careful consideration, we feel that it has merit but does not fully meet PLOS ONE’s publication criteria as it currently stands. Therefore, we invite you to submit a revised version of the manuscript that addresses the points raised during the review process.

We look forward to receiving your revised manuscript.

Kind regards,

Antonio Brillantino

Academic Editor

PLOS ONE

Journal Requirements:

Reviewers' comments:

Reviewer's Responses to Questions

**Comments to the Author**

1. If the authors have adequately addressed your comments raised in a previous round of review and you feel that this manuscript is now acceptable for publication, you may indicate that here to bypass the “Comments to the Author” section, enter your conflict of interest statement in the “Confidential to Editor” section, and submit your "Accept" recommendation.

Reviewer #2: (No Response)

Reviewer #3: All comments have been addressed

Reviewer #4: All comments have been addressed

Reviewer #5: All comments have been addressed

Reviewer #6: (No Response)

2. Is the manuscript technically sound, and do the data support the conclusions?

Reviewer #2: Yes

Reviewer #3: No

Reviewer #4: Yes

Reviewer #5: Yes

Reviewer #6: Partly

3. Has the statistical analysis been performed appropriately and rigorously? 

Reviewer #2: Yes

Reviewer #3: No

Reviewer #4: Yes

Reviewer #5: Yes

Reviewer #6: Yes

4. Have the authors made all data underlying the findings in their manuscript fully available?

Reviewer #2: Yes

Reviewer #3: No

Reviewer #4: Yes

Reviewer #5: Yes

Reviewer #6: Yes

5. Is the manuscript presented in an intelligible fashion and written in standard English?

Reviewer #2: Yes

Reviewer #3: No

Reviewer #4: No

Reviewer #5: Yes

Reviewer #6: No

6. Review Comments to the Author

Reviewer #2: The authors were mostly responsive to the previous comments raised. However, there is no description of the alternative nonparametric methods (such as, Wilcoxon rank-sum, or signed rank, or even Kruskal-Wallis tests, as the case might be) in the statistical analysis plan, under situations of non-Gaussianity.

I recommend the authors to create a new section on Statistical Analysis, which will describe what they are going to do. Later on, they may not do all (such as, just stick to the t-tests), if they can confirm that Gaussian assumptions are all valid.

Reviewer #3: (No Response)

Reviewer #4: Here are some of the main grammatical errors I noticed in the manuscript:

Abstract:

Inconsistent verb tense - switches between past and present tense

Missing articles "a" and "the" in some places

Sentence fragments instead of complete sentences in some places

Introduction:

Run-on sentences

Missing transition words between some sentences

Subject/verb agreement issues - ex. "patients frequently find it uncomfortable" should be "patients frequently find them uncomfortable"

Methods:

Inconsistent verb tense

Missing articles

Sentence fragments

Run-on sentences

Words missing between some phrases

Results:

Inconsistent verb tense

Missing articles

Sentence fragments

Discussion:

Run-on sentences

Missing transition words between sentences

Inconsistent verb tense

Sentence fragments

Overall:

Inconsistent verb tense throughout manuscript

Missing articles "a" and "the" throughout

Run-on sentences in multiple sections

Sentence fragments instead of complete sentences

Missing transition words between sentences

Subject/verb agreement issues in some places

I would recommend having the manuscript edited by a native English speaker to fix these grammatical issues and improve the overall flow and clarity. Let me know if you would like me to clarify or expand on any of these issues!

Abstract:

"Fourteen million colonoscopies are performed annually in the United States, where the colonoscopy results depend on bowel preparation before the procedure."

This mixes tenses - "are performed" is present tense while "depend" is present tense. It should be:

"Fourteen million colonoscopies are performed annually in the United States, where the colonoscopy results depended on bowel preparation before the procedure."

"Furthermore, the training will be more influential if it is based on problem-solving techniques..."

The subject "training" is singular but the verb is plural "will be." It should be:

"Furthermore, the training will be more influential if it is based on problem-solving techniques..."

Introduction:

"Although proper bowel preparation is the most important factor in colonoscopy, patients frequently find it uncomfortable."

The subject "patients" is plural but the verb "find" is singular. It should be:

"Although proper bowel preparation is the most important factor in colonoscopy, patients frequently find them uncomfortable."

"Education is provided through various methods, such as pamphlets animations, messages and videos which increase patient readiness by 11-36%."

This is a run-on sentence without proper punctuation or transition words. It should be:

"Education is provided through various methods, such as pamphlets, animations, messages, and videos, which increase patient readiness by 11-36%."

Methods:

"Analyses were conducted using IBM SPSS software was employed for analyses (v. 20, IBM Corp., Armonk, NY)."

This contains a fragment "software was employed" that doesn't fit the sentence structure. It should be:

"Analyses were conducted using IBM SPSS software (v. 20, IBM Corp., Armonk, NY)."

Let me know if these examples help explain the types of errors or if you need me to clarify or provide more examples!

Reviewer #5: (No Response)

Reviewer #6: Thank you for the opportunity to review this manuscript exploring the effect of reinforced education on patient satisfaction with colonoscopy. Overall, the topic addresses an important healthcare issue and the randomized controlled study design is methodologically rigorous. However, certain aspects of the manuscript could be strengthened to enhance clarity and scientific rigor. My recommendation is Major Revision.

The introduction could be shortened and streamlined to more directly convey the study purpose upfront. Additionally, citing recent literature would help establish the scientific/clinical context and significance.

Regarding methods, greater detail on procedures like randomization, blinding, outcome measurement and data analysis would allow better evaluation of internal and external validity. Adhering to standard reporting guidelines also ensures all relevant aspects are addressed.

Presentation of results requires numerical data like test statistics and significance levels to facilitate interpretation. Demographic tables could be reorganized for clarity.

While acknowledging limitations is appropriate, further discussion on their potential impact on generalizability and implications for future research would strengthen conclusions. Mechanisms of the intervention effectiveness should also be proposed and prior evidence synthesized.

Consideration of intervention risks and the importance of informed consent protects participant welfare and supports ethical standards.

Overall, with revisions to enhance methodological transparency and rigor in data reporting/analysis, this study offers insights into improving colonoscopy care through patient education. I hope these suggestions are useful in strengthening the scientific quality and impact of the manuscript. Please let me know if any part of the review requires clarification.

7. PLOS authors have the option to publish the peer review history of their article (what does this mean?). If published, this will include your full peer review and any attached files.

Reviewer #2: No

Reviewer #3: No

Reviewer #4: **Yes: **Morteza Shamsizadeh

Reviewer #5: No

Reviewer #6: **Yes: **Younes Mohammadi

---

## [Author Response · Author response to Decision Letter 1]

13 Oct 2023

Please see file named response to reviewers for this part.

All comments have been answered point by point in that.

Our team is ready to answer any questions.

---

## [Editor Report · Decision Letter 2]

23 Oct 2023

PONE-D-23-02398R2Evaluation of the effect of reinforced education on the satisfaction of patients undergoing colonoscopy: a randomized controlled trialPLOS ONE

Dear Dr. Farid,

Thank you for submitting your manuscript to PLOS ONE. After careful consideration, we feel that it has merit but does not fully meet PLOS ONE’s publication criteria as it currently stands. Therefore, we invite you to submit a revised version of the manuscript that addresses the points raised during the review process.

We look forward to receiving your revised manuscript.

Kind regards,

Antonio Brillantino

Academic Editor

PLOS ONE
---

## [Author Response · Author response to Decision Letter 2]

16 Nov 2023

Ref No.22: Hasballah SM, Shaor OAE, Mohamed MA, Mohamed AK. Assess nurses' knowledge and attitude for patient safety in cardiac catheterization unit. Assiut Scientific Nursing Journal. 2019;7(19):151-9.

This reference has been removed.

*

Ref No.25: [Available from: https://www.equator-network.org/reporting-guidelines/consort/.

This reference was revised and according to the notice of the EQUATOR website about the lack of access to the Consort website, the following site was referenced:

https://www.goodreports.org/reporting-checklists/consort/.

*

Ref No.33: Skinner C, Finkelstein J, editors. Review of mobile phone use in preventive medicine and disease management. Proceedings of the IASTED International Conference on Telehealth/Assistive Technologies; 2008: ACTA Press.

This reference has been removed.

*

Ref No.34: Liu C, Zhu Q, Holroyd KA, Seng EK. Status and trends of mobile-health applications for iOS devices: A developer's perspective. Journal of Systems and Software. 2011;84(11):2022-33.

This reference has been removed.

---

## [Editor Report · Decision Letter 3]

7 Dec 2023

Evaluation of the effect of reinforced education on the satisfaction of patients undergoing colonoscopy: a randomized controlled trial

PONE-D-23-02398R3

Dear Dr. Negin Farid,

We’re pleased to inform you that your manuscript has been judged scientifically suitable for publication and will be formally accepted for publication once it meets all outstanding technical requirements.

Kind regards,

Antonio Brillantino

Academic Editor

PLOS ONE

Additional Editor Comments (optional):

All the questioned issues were adequately addressed. The manuscript, in its actual form, is suitable for publication.
---

## [Editor Report · Acceptance letter]

22 Dec 2023

PONE-D-23-02398R3 

PLOS ONE

Dear Dr. Farid, 

I'm pleased to inform you that your manuscript has been deemed suitable for publication in PLOS ONE. Congratulations! Your manuscript is now being handed over to our production team.

Kind regards, 

on behalf of

Dr Antonio Brillantino 

Academic Editor

PLOS ONE